# Scale and information-processing thresholds in Holocene social evolution

Jaeweon Shin [1], Michael Holton Price[2], David H. Wolpert [2,3 ✉], Hajime Shimao[2], Brendan Tracey[2] & Timothy A. Kohler [2,4,5,6 ✉]

Throughout the Holocene, societies developed additional layers of administration and more information-rich instruments for managing and recording transactions and events as they grew in population and territory. Yet, while such increases seem inevitable, they are not. Here we use the Seshat database to investigate the development of hundreds of polities, from multiple continents, over thousands of years. We find that sociopolitical development is dominated first by growth in polity scale, then by improvements in information processing and economic systems, and then by further increases in scale. We thus define a Scale Threshold for societies, beyond which growth in information processing becomes paramount, and an Information Threshold, which once crossed facilitates additional growth in scale. Polities diverge in socio-political features below the Information Threshold, but reconverge beyond it. We suggest an explanation for the evolutionary divergence between Old and New World polities based on phased growth in scale and information processing. We also suggest a mechanism to help explain social collapses with no evident external causes.

[1] Department of Mathematics, Rice University, 6100 Main St, Houston, TX 77005, USA. [2] Santa Fe Institute, 1399 Hyde Park Rd, Santa Fe, NM 87501, USA. [3] Center for Biosocial Complex Systems, Arizona State University, Tempe, AZ 85281, USA. [4] Department of Anthropology, Washington State University, Pullman, WA 99164-4910, USA. [5] Crow Canyon Archaeological Center, 23390 C R K, Cortez, CO 81321, USA. [6] Research Institute for Humanity and Nature, 457-4 Kamigamo Motoyama, Kita-ku, Kyoto 603-8047, Japan. ✉email: dhw@santafe.edu; tako@wsu.edu

At any single moment over the last 10,000 years, there has been great variation in how the human societies of the world are organized. Despite this cross-sectional heterogeneity, striking regularities appear to govern the evolution of societies through time. The domestication of plants and animals and formation of sedentary agricultural villages developed independently in about a dozen separate regions, in both the Old and New Worlds, following climatic stabilization at the end of the last Ice Age[1–3]. In many of these societies, domestication of plants and animals combined with changing human mobility patterns to generate rapid population growth[4,5]. Rather similar processes of urbanization eventually followed this population growth in many parts of both the Old and New Worlds[6–8]. In most societies, wealth inequality increased with greater reliance on food production, with the development of technologies yielding more surplus, with more efficient wealth transmission between generations, and with more prominent political hierarchies and various other factors promoting economic defensibility[9–11].

More quantitatively, there is pronounced covariation among the values of many variables measuring various aspects of society when assessed over long periods. Productive agricultural systems, large group sizes, urbanism, political hierarchy, and high levels of wealth inequality often—though not completely—coincide. In every region of the world permitting agriculture, we find considerable similarity in both the beginning forms of human social groups, as relatively small-scale hunter-gatherer groups, and their ultimate (or at least current) forms, as large-scale urban societies. All the interesting variability—the history—lies in the differing ways that the societies in each region develop from those beginnings.

To determine whether these shared patterns in the dynamics of societies through time reflect propinquity and borrowing, shared roots, or entrainment along common paths via competitive pressures requires detailed data on their development around the world, stretching back at least to the end of the Pleistocene. Fortunately, our knowledge of Holocene prehistory and history has expanded rapidly in the last several decades, facilitated by the requirement in many countries for site investigation prior to destruction by development[12], by increased chronological accuracy (e.g., ref. [13]), and by improvements in the scope and precision of various climatic proxies that permit better understanding of the role of climatic variability in sociopolitical change[14]. Due to the rapid accumulation of such data, once-authoritative comparative studies (e.g., refs. [15,16]) deserve continuous re-evaluation, especially since these earlier studies emphasized the more recent, historical end of the spectrum of increasing sociopolitical complexity.

One notable project addressing this demand has resulted in a dataset called "Seshat: Global History Databank"[17]. This is an ambitious coding of more than 1500 variables containing state-of-the-art knowledge on variables related to social complexity for hundreds of polities. The polities in the Seshat database for which comprehensive data are available span six continents and date from the Neolithic to the middle of the last millennium.

This dataset has enabled a number of recent publications[17–21] (see also Supplementary Note 1: Overview of Seshat and Supplementary Note 2: Previous Seshat Research). Particularly important here, Peter Turchin and colleagues[22] combined 51 variables recorded in Seshat to produce nine new variables, which they call Complexity Characteristics (CCs). They then performed a principal component analysis (PCA) of 285 polities (as of March 25, 2020, the Seshat database contains 291 polities) in that nine-dimensional CC space, as described in more detail below. They found that the first principal component (PC/PC1) captures 77% of the underlying variation in the dataset. In addition, they found that polities almost always increase their PC1 value with

time. This led them to portray sociopolitical evolution as a largely homogeneous process through time in which scalar growth (in population, capital population, and territory size) and increase in the number of levels and sophistication of government, infrastructure, writing, texts, and exchange media all occur in a highly correlated fashion.

The PCA of Turchin and colleagues answers the question, "what is the largest dimension of shared variability across all the variables and among all the cases in the Seshat data?" where these cases range from village-level societies to empires. It is important to note that this question is essentially ahistorical, not only disembedding the cases (a particular polity in a particular century) from their regions, but also from their own trajectories of development, and from that of similar polities. In contrast, here we are more interested in questions like "how do polities become more complex as they evolve?" and "what paths do polities follow through the Seshat feature space?"

However, the fact that the first principal component of a PCA by itself captures a large amount of the large-scale variation in the data does not preclude there being rich structure relating the components of the (vector-valued) data at a more fine-grained scale. To give a stark example, suppose that we have a two-dimensional dataset, generated by sampling the two equations $x(t) = t$, $y(t) = \sin(t)$. (So $t$ is a hidden variable, not recorded in the data.) If the data come from a long-enough range of $t$ values, then a PCA will tell us that PC1 is (approximately) identical to the $x$ direction, while PC2 is identical to the $y$ direction, and that PC1 "captures a large amount of the variation in the data". Yet as shown in Fig. 1, in fact there is far more to understand about the relationship between PC1 and PC2, namely that $y = \sin(x)$ exactly. Ironically, the fundamental issue we raise is very closely related to one identified by Turchin and Korotayev who noted that correlational analyses of cross-cultural data yield misleading results when two variables (population density and incidence of warfare, in their case) are linked in a dynamical system and oscillate relative to each other with time lags, as in a predator-prey system[23]. In the case of the correlational analysis in Turchin et al.[22], the issue is not with time lags, though, but with changes in

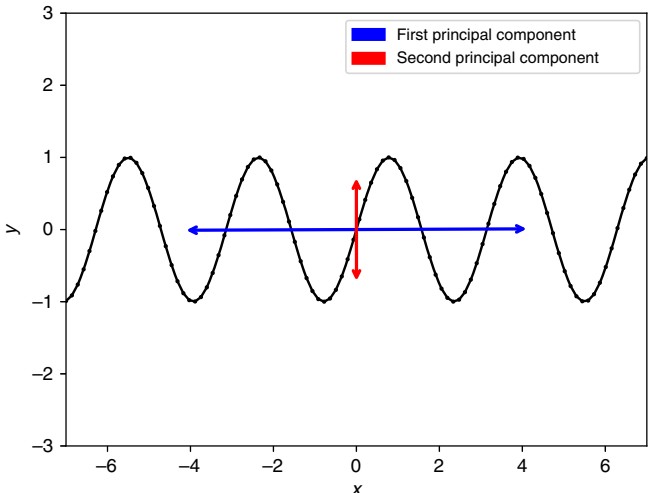

**Fig. 1 PCA decomposition of the data generated from sinusoidal function.** Datapoints on the plot are drawn at uniform interval from parametric equations $x(t) = t$ and $y(t) = \sin(2t)$, where $t$ ranges from −7 to 7. The blue bidirectional line represents the first Principal Component, which explains 98% of the variance in the data. The red bidirectional line represents the second Principal Component, which explains the rest of the variance in the data.

the direction of the relationship between PC1 and PC2 across the developmental ranges of the societies they describe.

Motivated by such considerations, in this article we analyze the covariation between the values of the top two principal components in the Seshat dataset. We find that there is a striking, highly patterned relationship between them—similar to the sine wave relationship of Fig. 1, in fact.

To help understand the social and historical significance of this relationship, we examine how the weightings (loadings) of the CCs under the vector PC1 and under the vector PC2 change as a polity develops. This provides our first result: According to the Seshat database, the development of polities on average is dominated first by a period of growth in scale, e.g., in the capital's population and territory size. After that the dynamics enters a period when it is dominated by improvements in information-processing and economic systems (i.e., by transactional and information-storage capabilities). These improvements are then followed by further increases in scale in a third period. This result suggests that to make major improvements in information processing, a polity must first surpass a scalar threshold. It also implies that the further increases in scale in the third region depend in turn on improvements in information-processing ability. Social evolution is thus a contingent, historical process, but one with strong constraints on its form imposed by structural considerations (in the sense of ref. [24]).

We interpret the two transitions between these three periods as permeable boundaries in social complexity. The first boundary moving from left to right along increasing values for PC1, which we call the Scale Threshold, separates those polities that have undergone more growth in scale than in information-processing capacity, from those polities that have already gone through such scale growth and are now differentially increasing their information-processing capacity. The second boundary, which we call the Information Threshold, separates polities that have not achieved the information-processing capacity that appears to be needed for additional growth in scale, and so are still growing that capacity, from those polities that have in fact achieved sufficient information-processing capacity for further increases in scale.

Next we investigate the dynamics of the NGAs recorded in the Seshat dataset, making full use of the century-resolution time-stamps provided in Seshat. This provides our second, albeit qualitative result: There appears to be a first region in PC1–PC2 space in which polities are relatively concentrated. This is followed by a second region where polities become more spread apart; we can speak of them taking different paths with respect to information processing as they grow in scale. In a final, third region, there appears to be a pronounced homogenization of features in which polities come to lie almost on top of one another. We note that this final convergence might simply reflect saturation of the values of the particular features that are recorded in Seshat, which were chosen with pre-modern polities in mind. Intriguingly, the first region roughly coincides with the scale-dominated growth described above, before the Scale Threshold is crossed. The second region coincides with a more variable dynamics, where many but not all polities achieve improvements in information-processing (in the sense discussed above). The third region roughly coincides with renewed emphasis on growth in scale, after the Information Threshold has been crossed.

While performing these analyses we uncovered patterns that turned out to be statistical artifacts. In particular, there is a pronounced clustering of the PC1 values in two distinct regions, which can be accurately described as a mixture of two Gaussians of very similar widths (see Supplementary Note 3: Bimodality and Supplementary Figs. 1–4). Such clusters in a sociopolitical feature space arise often in sets of data on long-term cultural evolution, and are sometimes interpreted as basins of attraction of an

underlying dynamics. However, as we note in the Supplementary Note 4 (Possible non-social-science explanations of the bimodality; see also Supplementary Figs. 5 and 6), the clustering in Seshat appears to be an artifact of a type not so far described in the literature, to our knowledge.

This provides a cautionary tale on analyzing datasets that combine samples of multiple, finite duration runs of the same Markov chain, if (as in Seshat) each run has a different starting position and a different starting time.

We now turn to presenting our results on the interaction between PC1 and PC2, with particular emphasis on how societies (henceforth, polities) move through the space jointly defined by these dimensions. We then discuss some implications of our findings for the role of population increase in social evolution, the divergence of polities in the Old and New Worlds, and possible new pathways for explaining social instability these results suggest.

## Results and discussion

**Interaction between PC1 and PC2.** We find a pronounced structure relating the values of PC1 with the values of the other eight principal components, despite PC1 containing 77% of the covariation across polity feature vectors. Here we report the relationship between the first and second principal components, PC1 and PC2. Figure 2 shows the average of polity values in PC2 evaluated in a sliding window along PC1 space. It is clear that a non-linear interaction between PC1 and PC2 exists, where the average PC2 scores of polities first drop, then rise, and then drop again as one moves along PC1. In other words, there is an interdependent relation between PC1 and PC2 despite no Pearson correlation in aggregate.

The first hinge point in Fig. 2 occurs near the PC1 value of −2.5 where average PC2 values abruptly stop decreasing and start to increase. Polities located near this point include Upper Egypt ca. 3600 BC, the Paris Basin ca. 400 BC (La Tène A), the Iceland Commonwealth ca. AD 1000, and the Valley of Oaxaca at the same time.

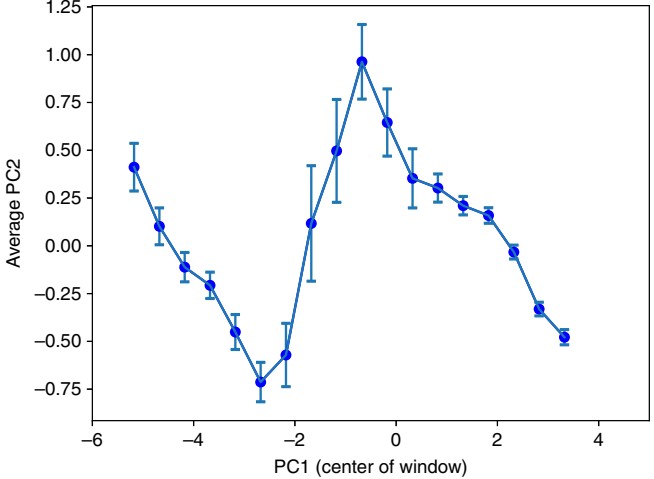

**Fig. 2 The average score of observations on PC2 in a sliding window along PC1.** The PCA is based on 414 datapoints. Each PC1 sliding window is defined to have width 1.0 (on the scale of PC1) and overlap width of 0.5 with other sliding windows, i.e., the center of the PC1 sliding window occurs at every 0.5 interval. The error bars are defined as the mean(PC2) ± SE (PC2), i.e., mean of the PC2 values within the sliding PC1 windows +/− standard error of the PC2 values. Number of samples in each sliding window, left to right, is 12, 25, 48, 50, 37, 35, 28, 20, 21, 22, 33, 35, 51, 89, 104, 86, 64, 42.

**Table 1 Loadings of the nine complexity components onto PC1 and PC2, and the percentage of variance of the full dataset explained by each PC. We present our calculations which differ very slightly from those presented in ref. [22].**

|  | PolPop | PolTerr | CapPop | levels | gvrnmt | infrastr | writing | texts | money | %Var |
|---|---|---|---|---|---|---|---|---|---|---|
| PC1 | 0.93 | 0.85 | 0.89 | 0.90 | 0.88 | 0.88 | 0.87 | 0.92 | 0.80 | 77.2 |
| PC2 | −0.25 | −0.35 | −0.27 | −0.15 | 0.09 | 0.12 | 0.31 | 0.24 | 0.28 | 6.0 |

The second hinge point in Fig. 2 occurs at a value for PC1 of around −0.5, where average values of PC2 suddenly switch from increasing to decreasing as PC1 increases. Polities located near this hinge include proto-Elamite Susiana ca. 3000 BC, and First Intermediate period Upper Egypt ca. 2100 BC.

It is important to be clear about how the changes in direction in Fig. 2 should be interpreted. Movement to the right along the PC1 axis happens as polities increase their values of those CCs that have positive loadings on PC1 while decreasing those that have negative loadings. Due to the choice of CCs in ref. [22] to be quantities that one would expect to increase together, they all have positive loadings on PC1 (though led most strongly by the CCs that the Seshat database labels *PolPop*, *texts*, *levels*, and *CapPop*—see top row in Table 1). Since PC2 must be orthogonal to PC1, this means that PC2 must both have CC components with positive loadings and those with negative loadings (see second row in Table 1). The movement downwards in PC2 on the left-hand side of Fig. 2 reflects increasing values on those CCs that have negative loadings in PC2—led by *PolTerr*, *CapPop*, and *PolPop*—while values on CCs with positive loadings in PC2 (led by *writing*, *money*, and *texts*) increase more slowly or not at all. On the other hand, movement upwards along PC2 happens when values for those CCs with positive loadings on PC2 are differentially increasing more than the others. Together these three CCs of *money*, *writing*, and *texts* with positive loadings (and less importantly those with lower positive scores on PC2 in Table 1) drive the changes in direction along the PC2 axis in Fig. 2. They reflect what we call the information-processing capacity of a polity.

These changes in which CCs drive the dynamics along PC2 are illustrated in Fig. 3, which decomposes PC2 into two sets of CCs, ones with positive loadings on PC2 (*money*, *writing*, *texts*, *infrastructure*, and *gvrnmt*) and others with negative loadings (*PolTerr*, *CapPop*, *PolPop*, and *levels*). Again, the CCs with positive PC2 loadings are related to information-processing capabilities, while those with negative loadings are related to scale. For each PC1 value, the red and blue lines represent the contribution of the two sets of CCs to PC2 (the sum of the product of the CC values and their loading on PC2). In other words, the red (blue) line indicates the PC2 value evaluated only using the information- (scale-) related CCs. The black line is the summation of the red and blue, and is equivalent to the line in Fig. 2.

As polities grow in their PC1 value, the relative contributions of the scale CCs and the information-processing CCs changes. Up to the first hinge point at PC1 ∼−2.5, polities grow substantially in scale as their PC1 value increases (i.e., the blue line drops), with relatively less growth in their information-processing capabilities. Since the PC2 vector has negative components for those scale-related CCs, this results in declining values on PC2. Between −2.5 and −0.5 on PC1, movement along PC1 is accompanied by relatively rapid increases in information-processing capabilities (though the error bars in Fig. 2 indicate more heterogeneity in scores on PC2 in this portion of PC1 than in any other portion). Since the PC2 vector has positive components for the information-processing CCs, this results in an increasing average score for polities on PC2 as PC1 grows in this region. Then as

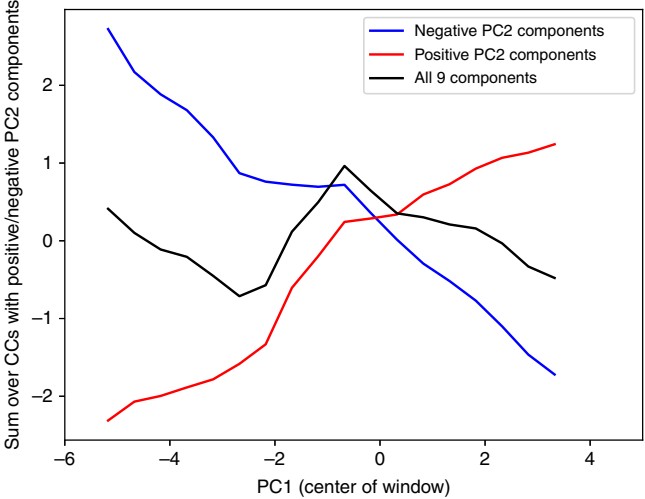

**Fig. 3 The decomposition of the average PC2 into two subsets of CCs.** The red line represents the CC components related to information-processing capabilities (*writing*, *money*, *texts*, *infrastructure*, and *gvrnmt* in order of importance), having positive correlations with PC2. The blue line represents the CC components primarily related to scale (*PolPop*, *levels*, *CapPop*, and *PolTerr* in order of importance). Note the changes in gradients along these two lines as they traverse PC1. The black line here is identical to the blue line in Fig. 2, with a change of scale.

polities move past −0.5 in PC1, again growth in scale CCs dominates the change in information-processing CCs, resulting in the decreasing values of PC2 in Fig. 2.

These dynamics suggest that at a level of complexity represented by values of PC1 of about −2.5, growth in scale may become constrained in polities where writing, money, or texts are relatively unsophisticated (store little information). Developing increased capacities in these areas appears to allow further increases in scale. This is why we suggest that that hinge point can be interpreted as a Scale Threshold marking an important transition in the dynamics of developing polities. At PC1 values of about −0.5 another, Information Threshold, is passed, appearing to facilitate further increases in scale.

**Dynamics in the PC1–PC2 space.** It is possible that the results in Figs. 2 and 3 could be driven by a few aberrant cases, perhaps a few large societies of the Americas that lack writing or some small societies that have writing for some reason like borrowing, or because they have budded off from a larger society. (In fact, all these possibilities were raised by a reviewer.) To help address such issues, in Fig. 4 we present a more fine-grained picture of the Seshat PC1–PC2 values, by including the time-stamps of the datapoints to depict trajectories of individual NGAs. Arrows run from an entry in Seshat giving the (PC1, PC2) coordinates of a particular polity at a particular time to the (PC1, PC2) coordinates of the next recorded time for that same polity.

Typically, the time between such datapoints is 100 years, but often there are gaps in the sequence[22]. For observations more than 100 years apart, the dots in Fig. 4 show the interpolated

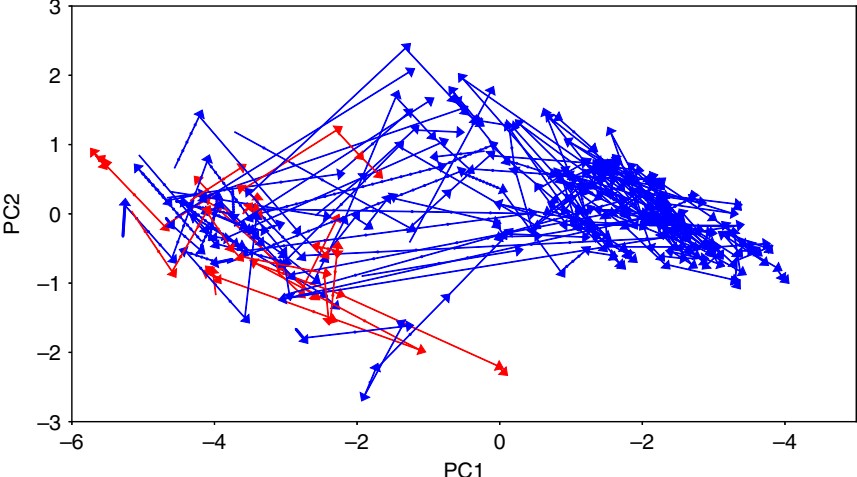

**Fig. 4 The trajectories of individual NGAs in PC1–PC2 space.** Red indicates New World NGAs; blue indicates Old World NGAs.

locations along each arrow at a spacing of 100 years, given a linear interpolation between the points $x_i$ and $x_{i+1}$. Red arrows stand for New World polities, and blue for those in the Old World.

Figure 4 suggests several features of sociopolitical evolution that have been previously poorly recognized (if at all). First, as mentioned above, the thresholds picked out by the hinge points in Fig. 2 approximately coincide with other changes in the dynamics through PC1–PC2 space. For example, societies lying between the Scale and the Information thresholds are more heterogeneous in their movement in both PCs than are societies lying to the right of the Information Threshold. Movement past that threshold, into the right-most region, seems to be in part a process of homogenization of features.

Second, the movement upward in the PC2 dimension between values of about −2.5 to near 0 on PC1 is nearly universal in the Old World regions analyzed in ref. [22]—though not in the Americas. (The six NGAs analyzed in the New World are the Finger Lakes, Cahokia, and Valley of Oaxaca in North America, and the Lowland Andes, North Colombia, and Cuzco in South America. The complete sample is listed in Turchin et al. [ref. [22], Table SI1]; see also Supplementary Fig. 10). Evidently, New World behavior is indeed unusual—but works to weaken the patterns in Figs. 2 and 3 rather than strengthen them. We leave for the future more work on the relationship between New World polities and the Scale Threshold, especially since the sample from Seshat analyzed here omits key New World societies, such as the Maya, Teotihuacan and the Aztec.

In the current sample, New World societies are also unusual in that they are largely confined to a loose left-hand cluster, centered around (P1 = −3.5, PC2 = −0.5). Cahokia, for example, achieves its highest score on PC1 at A.D. 1200 (−2.57; PC2 = −1.21), approaching but not quite crossing the Scale Threshold. In this sample, only the Cuzco NGA under the Inca crosses the Information Threshold, during the AD 1400s and 1500s, with a PC1 score of 0.02 in A.D. 1500, but markedly low scores on PC2 (−2.26 in A.D. 1500). For the moment we conclude that societies in the New World sample may be constrained in size by their low information-processing capacities.

Perhaps the most remarkable feature in Fig. 4 though is the nearly universal movement upwards in the PC2 dimension in the middle of the PC1 range, in between the two thresholds (especially if one restricts attention to Old World societies). This movement is what causes the change in direction of the line in Fig. 2 (To reiterate the point made above, this reflects differential

improvements in information processing in the middle zone of PC1 than in either of the other two zones.).

The general downwards slope of most trajectories for positive values of PC1, as polities pass the Information Threshold, is also noteworthy. In the part of the space with even greater PC1 values it is tempting to conclude from informal examination of the plot, and its dense cluster of points around (PC1 = 2, PC2 = 0), that there is a sink in the middle of the right cluster. While possible, such a cluster would need to be confirmed with further statistical analysis.

A final point of interest demonstrated in Fig. 4 is that human societies do not exist in most of the possible space of joint social and cultural traits. The later cluster is even more focused on a portion of the available space than are the societies in the looser, earlier cluster. We do not (for example) see societies with sophisticated information processing but small size. This suggests that it is impossible for human societies to arise (let alone persist) unless they satisfy a tight set of constraints limiting their sociopolitical characteristics.

**Thesholds and morphospace diversity.** To a first approximation, the Seshat database analyzed here and in ref. [22] demonstrates a shared movement through time by all the polities in this sample along an axis (PC1) representing simultaneous increases in scale and in a number of indicators of governmental, infrastructural, informational, and economic capacities. This homogeneous process is characterized as having produced a "striking similarity in the way that the societies in our global historical sample are organized" ([ref. [20], p. 4 of 8]; see also [ref. [22], p. 3]).

Going beyond this first approximation, however, we demonstrate a pronounced structure relating the scale and information complexity characteristics recorded in Seshat. This structure strongly suggests that sociopolitical evolution is initiated primarily by growth in scale. Only once a certain size is reached, which we call the Scale Threshold, does growth in information-processing capacity start to accelerate relative to growth in scale. This suggests that a polity's growth in scale may become limited by low capacity in its writing, exchange and similar information-processing systems. Once those capacities reach a certain level, polities can cross what we call the Information Threshold, allowing further growth in scale. We emphasize of course that these thresholds are not Iron Curtains separating incommensurable worlds, but elastic and porous frontiers between zones in which different constraints are most pronounced.

Moreover, the leftmost zone in Figs. 2–4, corresponding to the first stage of this dynamics, seems to be one in which different polities develop in idiosyncratic ways, diverging in the space of sociopolitical features. The second stage, dominated by growth in information-processing capabilities, seems to be one in which polities—with the notable exception of those in the New World— all start moving in parallel. Then in the third stage—across the Information Threshold—polities converge in feature space. We caution that these patterns are all based on the current Seshat database which, impressive as it is, is still quite small.

Peregrine[25] reports an analysis of the archaeological traditions coded in ref. [26] in which factor analysis was used to define a two-dimensional morphospace within which each of the traditions could be located at a specific time. As in our analysis of Seshat, a great deal of the available morphospace was never (or rarely) occupied. Following ref. [27], Peregrine attributes some of this empty space to functional constraints (for example, were small-scale societies to undertake construction of large-scale public monuments, it could only be to the [lethal] detriment of their day-to-day food-getting activities). Other empty space is likely due to developmental constraints (for example, small-scale societies may simply not have enough interacting minds to innovate and maintain complex technologies).

The observation that societies focus on a smaller portion of the possible space defined by PCs 1 and 2 as they move into the right-most cluster is reminiscent of the common observation that small-scale societies are relatively more diverse than large-scale societies in terms of both cultural practices and organization (e.g., [ref. 28, pp. 8–10]). Note in particular that in the part of the transition region where scores for PC1 hover around 0, polities with low scores on PC2 tend to move up, while polities with high values on the second PC tend to move down (Fig. 4). A plausible mechanism for this convergence that deserves further examination is that the selective environments inhabited by societies become increasingly competitive as they move to the right along the PC1 dimension. Highly selective environments reward societies with constellations of traits that allow them to survive and spread. Diverse societies either develop similar constellations of traits through this selection process, or they fail to survive. Similar dynamics are predicted in refs. [29,30].

Three points raised by our analysis warrant further discussion. The first has to do with how social evolution has typically been explained. The second concerns the Old/New World divergence revealed here. Third, this analysis points to new avenues for understanding social instability. We address these in turn below.

**Role of population**. Archaeologists have long held different views on the importance of population growth in social evolution. From a perspective grounded in the small- and middle-range societies in the US Southwest, for example, Kohler and Crabtree[31] have recently argued that increasing social and political complexity is largely driven by population growth. From a point of view more connected with the ancient states of Mesoamerica, however, Feinman[32] recognizes relatively strong correlations between population size and sociopolitical complexity, but terms the relationship "messy," arguing that it is significantly mediated by the "specific nature of the relational links between individuals" especially the specific organization of the institutions by which different societies are governed [ref. 32, p. 47]. Our findings here —that growth in scale is dominant at the extrema of PC1 but that development of information processing and exchange mechanisms is critical in the middle-range of PC1—suggest that both of these perspectives can be accommodated.

Our results here are in general accord with the findings of an earlier body of research in anthropology on scalar stress that

modeled a relationship between increasing group size and the appearance of hierarchical structures facilitating information processing[33,34]. Such research provides the uniformitarian microfoundations for our own work, via studies of small groups, variability in spans of control in organizations of various types, and ethnographies. We focus instead on understanding how this relationship played out in the global history of Holocene societies.

Moreover, our analysis raises the possibility that the zone between the thresholds of Scale and Information is especially critical for the evolution and elaboration of institutions. Institutions are mechanisms "whose outcomes are rules of interactions" and they "provide a means for individuals to amalgamate dispersed information about resources and wants, and hence coordinate … actions to reach an equilibrium that gives higher pay-offs" than would be possible in their absence [ref. 35, p. 3–4]. Institutions maintain and promulgate information about peoples' past behaviors—information that becomes more difficult to recover as group sizes grow [ref. 35, p. 4]. So institutions thrive on and deal in information, and the actors controlling them will seek to improve the mechanisms for its storage and retrieval.

For example, Koji Mizoguchi describes Japan's Kansai region ca. A.D. 400 (Middle Kofun, PC1 score −0.57) as experiencing an increasing number of distinct fields of communication resulting in part from extension of political interests into neighboring regions of Korea, coupled with attempts to institutionalize communications. By A.D. 500 (late Kofun, PC1 score 0.20) this region has crossed the Information Threshold into an "archaeology of bureaucracy" in which the elite were paving the way for "registration and direct control of commoners" [ref. 36, p. 308]. Within a century, by A.D. 600 (Asuka, PC1 score 1.64) the Kansai region hosted the rise of the palace, the introduction of Buddhism, the first national treasury, the introduction of Chinese-modeled fiscal policies and script, and formalization of the *Ritsuryō* law code. The fifth and sixth centuries likewise saw significant spatial expansion in the area controlled by the dominant clan, presumably enabled, in part, by these advances.

**Diverging old and New World trajectories**. One unexpected finding is that few New World societies seem to cross the Scale Threshold. We infer that as a result they do not undergo as much pressure to develop information and exchange systems as societies that did. This suggests that if we want to understand the rarity of writing systems in the New World and the absence of coinage we first need to understand the factors limiting scale in many such societies. There are many candidates, including perhaps the absence in the New World of an inland sea providing an efficient means of linking societies on its periphery.

More promising as an explanatory factor though in our view is the general absence of animals capable of carrying people or sizeable loads in the Americas. This is also of course connected with the famous absence of the wheel as a practical device. Such animals, of course, dramatically reduce the friction of transport and facilitate expansion of empire. It is notable that the only partial exception in the Seshat sample is the Cuzco NGA where camelids could carry loads, but not warriors. Not coincidentally we think, this is also the only New World society in this sample that crosses the Scale Threshold, and perhaps as a result the only to develop a recording system (the *quipu*) beyond mnemonic devices.

In view of these results, the question "Why didn't indigenous North American societies develop writing systems?" seems ill-posed. The appropriate question is perhaps "Why didn't these societies develop in scale to the degree where writing systems would have been advantageous or required?"

**Explaining social instability**. In most of the polities recorded in Seshat, there is a strong relationship between movement along the PC1 dimension (growth in scale coupled to growth of information-processing capacities)—but not in all of them. Could some of the frequent collapses seen in societies be due to a polity's never developing sufficient information-processing capacities, so that it stumbles or even collapses through poor performance due to lack of external connectivity, internal coherence, or inability to compete with polities whose superior information-processing abilities have enabled more growth in size? While Seshat is currently too small to address whether such "correlational failures" should be taken seriously, that may not be the case in the future.

Archaeologists have long noted the instability of societies of intermediate complexity. The Mississippian (late-prehistoric, maize-based) societies of the US Midwest and Southeast, traditionally characterized as simple or complex chiefdoms, commonly cycled unstably between these two organizational forms[37] or, in a related vision, oscillated between more dispersed and more concentrated power distributions in a near-constant churn of fission-fusion[38]. Proximate mechanisms driving this instability appear to have included both endogenous and exogenous factors: contention over chiefly succession, the fortunes of war, and climatically induced variability in the maize that fueled their tribute-based hierarchies. While such factors may in fact be sufficient, the results here suggest the further possibility that mismatch among aspects of social development should be considered as a generic cause of instability.

For example, Fried [ref. 39, p. 225] surmised that stratified societies lacking state institutions must be "one of the least stable models of organization that has ever existed" since he thought that stratified communities would have to quickly develop ever more powerful institutions of political control to maintain the differential access to basic resources that defines systems of stratification. Analyzing such problems as failures of correlation among the sociopolitical features of a polity might provide a more general view on the nature of collapses without clear exogenous triggers. Pursuing such a vision is a topic for future research.

## Methods

**Sample**. We analyze a worldwide sample of 285 polities in 30 regions described on 51 variables relevant to sociopolitical organization by reference to archaeological and historical data, using the codings by century derived from and made available by Seshat: The Global History Project[17] (Supplementary Note 1: Overview of Seshat). Codings for each region begin as early in the Neolithic as permitted by the local archaeological record and extend through time up to just before the local industrial revolution. These 51 variables were collapsed into nine Complexity Characteristics (CCs) named in Table 1 by Turchin et al.[22] whose usage we adopt. Four of these measure aspects of social scale: Polity Population, Polity Territory, Capitol Population, and Hierarchy (i.e., number of levels in the political, military and religious establishments, and in the settlement hierarchy). The remainder either measure how information and financial transactions were processed (writing, money) or mix aspects of scale and information. Multiple imputation was used to deal with missing data, uncertainty, and expert disagreement. Turchin et al.[22] demonstrated that a first component (PC1) from a principal component analysis (PCA) explains ~77% of the variability in this dataset whereas PC2 explains ~6%; the remaining components drop rapidly in explanatory value. Turchin et al.[22] restrict their discussion of the PCA to the first principal component. Our analyses here do not overlap with more recent analyses by the Seshat team (Supplementary Note 2: Previous Seshat Research).

**Analysis**. We reanalyze the dataset reported by ref. 22, accepting their PCA but extending our analysis to the first two PCs. Two facts motivate and justify this extension. First, their claims about the proportions of variance explained by each PC are based on the static pattern in the data and do not take into account the dynamic development of this pattern through time, which involves moving through a non-linear relationship between PCs 1 and 2 (Fig. 2). Second, the scores on PC1 in their analysis are bimodal, whereas strict interpretation of percentages explained by each component requires unimodality[40].

We investigate this bimodality extensively in Supplementary Note 3: Bimodality. We use Gaussian mixture models to show that there are two discrete clusters in the original 9-dimensional data, and we demonstrate via bootstrapping

that these are not caused by noise. We develop a novel model explained in detail in Supplementary Note 4.1: Discrete Markov Transition Model that on the whole suggests that the mode consisting of low values on PC1 is likely to be due to non-uniformity in initial values on CCs (and hence on PC1) when polities first begin to be measured by Seshat researchers. We develop some simple simulations to help judge how likely it is that the right-hand mode in the distributions of polity scores on PC1 is due to saturation in values of the variables underlying the PC. These simulations show that if saturation were responsible for the right-hand peak, this peak should occur at the positive limit in Supplementary Fig. 1, instead of at lower values of PC1.

Another way in which bimodality in the distribution of polity PC1 values could arise would be if polities between the two modes were undersampled (i.e., via selection bias). We investigate this in the Supplementary Note 4.3: Undersampling by (1) interpolating missing time periods within each region; and (2) weighting within each region depending on the sparsity of data in the neighborhood of each data point. Since bimodality survives both tests, we tentatively conclude that it is not due to selection bias.

Our main results are derived from visual analysis and interpretation of Figs. 2 and 3, and especially the dynamics displayed in Fig. 4.

**Reporting summary**. Further information on research design is available in the Nature Research Reporting Summary linked to this article.

## Data availability

The data that support the findings of this study are available through https://github.com/jaewshin/Holocene. We used data from the Seshat databank (seshatdatabank.info) under Creative Commons Attribution Non-Commercial (CC By-NC SA) licensing (https://creativecommons.org/licenses/by-nc-sa/4.0/legalcode). The data were accessed and downloaded in August 2018 and are identical to the dataset published and used by Turchin et al.[22]. The data contain 285 unique polities and have a total of 414 datapoints. An updated version of the dataset, which contains 291 unique polities and 864 datapoints, is available for download through http://seshatdatabank.info/datasets/.

## Code availability

The code for all figures and analyses is available through https://github.com/jaewshin/Holocene.

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

## Acknowledgements
This material is based upon work supported by the National Science Foundation under Grant No. SMA-1620462 to D.H.W and T.A.K. We thank Darcy Bird, Laura Ellyson, Peter Turchin and the Seshat project for providing the dataset used here, Henry Wright, and the Santa Fe Institute and the Research Institute for Humanity and Nature for support.

## Author contributions
D.H.W, H.S., M.A.P., and T.A.K. designed the research. T.A.K., D.H.W., and M.H.P. wrote the main text. J.S., H.S., B.T., M.H.P., D.H.W., and T.A.K. wrote the SI. J.S., H.S., M.H.P., and B.T. conducted the analyses and prepared the figures.

## Competing interests
The authors declare no competing interests.
