## [Peer Review File · Nature Communications]

Reviewers' Comments:

Reviewer #1:

Remarks to the Author:

This is an excellent paper that deserves publication in Nature Communications with only minor changes. The authors expand upon work recently published in PNAS that used PCA to argue that change in population (social scale) was the major driver of human cultural evolution. This paper develops that finding and looks at other components manifest in the underlying data and their interaction with population. The authors find that capacity for information processing is also a key driver of cultural evolution, and argue that this finding helps to explain the interesting pattern of punctuated equilibrium that many scholars have identified in cultural evolutionary sequences. The authors also provide a brief counter-argument to an article on the evolution of moralizing high gods that recently appeared in Nature.

The paper's content and findings are novel and will be of interest to a broad audience of scholars in the historical and evolutionary sciences. This paper fits within the interdisciplinary focus of Nature Communications, but might be more appropriately placed in Nature Human Behavior or Nature Ecology and Evolution (where a commentary on the importance of studying cultural evolution appeared in one of the early issues). In any case, this is an important paper and is certainly of the quality and impact to be published in Nature Communications.

The data used by the authors is based on the Seshat databank and have already proven themselves to be useful in exploring human cultural evolution. They are robust data, valid and reliable, and completely appropriate to the author's purpose. The analyses performed on these data are clear, appropriate, and well-described. The interpretation made of those findings are reasonable and clearly presented.

I am a bit puzzled about the extensive analyses of bimodality presented in the supplementary materials, as they seem unnecessary in terms of supporting the specific analyses presented in the paper. They are quite interesting, however, and I would urge the authors to consider publishing them separately.

I also do not think that the short critique of the moralizing high gods paper by Whitehouse et al. is necessary. The section does not add much to the paper's core analyses or argument, and to me only distracts from the paper's overall flow. I would suggest leaving it out.

The authors do a good job in the short space they have discussing relevant literature, but I would suggest they give a nod to Greg Johnson and to Krisztina Kosse who built the foundation upon which the general "scalar stress" interpretation

of the relationship between population, information, and cultural change was built.

In sum, this is an excellent paper with potentially broad impact and appeal and deserves publication.

Peter Neal Peregrine

Reviewer #2:

Remarks to the Author:

This manuscript addresses the fundamental question: What is the relationship between the increase in size of a polity and the increase in its information processing capabilities? Is an overall increase in the social complexity of a polity (e.g. number of levels of hierarchy) driven first by an increase in polity size, or first by an increase in information processing capabilities and institutions such as money? For the first time, this manuscript places these in temporal order when considering data from a large number of polities around the world. By building on a previous Principle Components Analysis conducted by Turchin and colleagues, which considers 9 complexity characteristics for 200 polities over time, the manuscript delineates between the two alternatives by considering (graphically) the relationship between the first and second principle components. The conclusions are non-trivial but also intuitive. They are that polities tend to first increase in size, until they reach a size threshold limit. Polities then need to increase their information processing capabilities to grow further. Finally, a limit in the amount that information processing capabilities can increase is reached, which is only crossed by further increases in size.

I find the analysis sound and am supportive of publication. I have minor changes to suggest:

1. Presumably the process of increase in size, then in information processing, then in size again is cyclic, with the second growth in size presumably allowing for a second increase in information processing capabilities. Would you expect to see this in the dataset if it was extended up to the 21st century? The industrial revolution could be ideal to consider in this context. However, your results also suggest there may be a sink point around a value of 2 for PC1, which would suggest the process cannot continue?
2. On page 2 it would be good to clarify exactly how many polities are in the Seshat database. 285 polities are used for the analysis, but what percentage of the database is this?

3. Page 3: "This is followed by a second region where polities become more spread apart; we can speak of them taking different paths in their growth in scale" -- I think this should be "different paths in their growth in information processing capabilities".

4. Page 5 mentions error bars. These should be shown on the graphs.

5. Page 5 states that growth in scale may be constrained by unsophisticated money or writing. It could also be constrained by "texts", which has a relatively high loading on PC2.

6. In the caption of Figure 3 it would be good to stress that the gradients of the lines are the key aspects that the reader should focus on.

7. On page 10 I find the claim that small-scale societies are relatively more diverse than large-scale societies surprising. Although you do provide a reference, it would be good to expand on the argument for this if possible.

8. When talking about institutions, be careful not to anthropomorphise too much. Under the definition of institutions you cite, an institution is defined by a political interaction that sets the rules for a subsequent economic interaction. This is different to the notion of institutions as actors, e.g. a university as an institution. In this sense, institutions themselves would not seek to improve mechanisms for information storage and retrieval. Rather, the actors in the political interaction (e.g. chiefs) would seek to do this, so they can better shape the rules of the economic interaction to benefit themselves.

9. SI page 1: "at most one NGA controlling the polity at any time", the orders of NGA and polity should be switched here.

REVIEWERS' COMMENTS:

Reviewer #1 (Remarks to the Author):

This is an excellent paper that deserves publication in Nature Communications with only minor changes. The authors expand upon work recently published in PNAS that used PCA to argue that change in population (social scale) was the major driver of human cultural evolution. This paper develops that finding and looks at other components manifest in the underlying data and their interaction with population. The authors find that capacity for information processing is also a key driver of cultural evolution, and argue that this finding helps to explain the interesting pattern of punctuated equilibrium that many scholars have identified in cultural evolutionary sequences. The authors also provide a brief counter-argument to an article on the evolution of moralizing high gods that recently appeared in Nature.

We thank the reviewer for these endorsements.

The paper's content and findings are novel and will be of interest to a broad audience of scholars in the historical and evolutionary sciences. This paper fits within the interdisciplinary focus of Nature Communications, but might be more appropriately placed in Nature Human Behavior or Nature Ecology and Evolution (where a commentary on the importance of studying cultural evolution appeared in one of the early issues). In any case, this is an important paper and is certainly of the quality and impact to be published in Nature Communications.

The data used by the authors is based on the Seshat databank and have already proven themselves to be useful in exploring human cultural evolution. They are robust data, valid and reliable, and completely appropriate to the author's purpose. The analyses performed on these data are clear, appropriate, and well-described. The interpretation made of those findings are reasonable and clearly presented.

I am a bit puzzled about the extensive analyses of bimodality presented in the supplementary materials, as they seem unnecessary in terms of supporting the specific analyses presented in the paper. They are quite interesting, however, and I would urge the authors to consider publishing them separately.

At the editor's suggestion, we retain the analyses of the bimodality (in the SI). This is also our strong preference. We point out that it is exactly that analysis which allows us to confidently respond to the comment by reviewer #2, concerning the apparent sink. Moreover, it is important for us to document the bimodality, as it is one of the reasons why we distrusted the original statement in Turchin et al. (2018) that PC2 was statistically insignificant.

I also do not think that the short critique of the moralizing high gods paper by Whitehouse et al. is necessary. The section does not add much to the paper's core analyses or argument, and to me only distracts from the paper's overall flow. I would suggest leaving it out.

Done.

The authors do a good job in the short space they have discussing relevant literature, but I would suggest they give a nod to Greg Johnson and to Krisztina Kosse who built the foundation upon which the general "scalar stress" interpretation of the relationship between population, information, and cultural change was built.

We thank the reviewer for this suggestion, which we wish we would have thought of ourselves. We have added a ¶ in the subsection “Role of Population” to this effect.

In sum, this is an excellent paper with potentially broad impact and appeal and deserves publication.

Peter Neal Peregrine

Reviewer #2 (Remarks to the Author):

This manuscript addresses the fundamental question: What is the relationship between the increase in size of a polity and the increase in its information processing capabilities? Is an overall increase in the social complexity of a polity (e.g. number of levels of hierarchy) driven first by an increase in polity size, or first by an increase in information processing capabilities and institutions such as money? For the first time, this manuscript places these in temporal order when considering data from a large number of polities around the world. By building on a previous Principle Components Analysis conducted by Turchin and colleagues, which considers 9 complexity characteristics for 200 polities over time, the manuscript delineates between the two alternatives by considering (graphically) the relationship between the first and second principle components. The conclusions are non-trivial but also intuitive. They are that polities tend to first increase in size, until they reach a size threshold limit. Polities then need to increase their information processing capabilities to grow further. Finally, a limit in the amount that information processing capabilities can increase is reached, which is only crossed by further increases in size.

I find the analysis sound and am supportive of publication. I have minor changes to suggest:

We thank the reviewer for this endorsement and for his/her understanding of our goals.

Presumably the process of increase in size, then in information processing, then in size again is cyclic, with the second growth in size presumably allowing for a second increase in information processing capabilities. Would you expect to see this in then dataset was extended up to the 21st century? The industrial revolution could be ideal to consider in this context.

While it is very tempting to speculate about how these trends might extend beyond the end of the data set, into the more recent past and even the future, here we are careful just to describe what the dataset says directly about the period it covers.

However, your results also suggest there may be a sink point around a value of 2 for PC1, which would suggest the process cannot continue?

We believe that within the context of the codings for the variables used in the Seshat dataset as it currently exists, this process could not continue. It is of course possible that in reality it does continue, but that would unfortunately be beyond the ability of the variables/values currently available to document, and we would rather not speculate as to what we might find with a dataset appropriate to this enlarged question.

On page 2 it would be good to clarify exactly how many polities are in the Seshat database. 285 polities are used for the analysis, but what percentage of the database is this?

Thanks, we clarify this now in the Introduction and also in the Data Availability section.

Page 3: "This is followed by a second region where polities become more spread apart; we can speak of them taking different paths in their growth in scale" -- I think this should be "different paths in their growth in information processing capabilities".

Well, it is really both simultaneously. We have adjusted the wording slightly here.

Page 5 mentions error bars. These should be shown on the graphs.

We meant that reference to be only to Figure 2, but unfortunately that wasn't clear; we have modified the text to make it clear. Figure 2 does have error bars, and we have added text to its legend to better explain what they indicate and how they were calculated.

Page 5 states that growth in scale may be constrained by unsophisticated money or writing. It is could also be constrained by "texts", which has a relatively high loading on PC2.

Good point; we have made this addition.

In the caption of Figure 3 it would be good to stress that the gradients of the lines are the key aspects that the reader should focus on.

We have added "Note the changes in gradients along these two lines as they traverse PC1" to the legend of Figure 3.

On page 10 I find the claim that small-scale societies are relatively more diverse than large-scale societies surprising. Although you do provide a reference, it would be good to expand on the argument for this if possible.

Since this is a minor point for this manuscript, we have elected not to follow this suggestion; developing this idea would be interesting but distracting from our main purpose.

When talking about institutions, be careful not to anthropomorphise too much. Under the definition of institutions you cite, an institution is defined by a political interaction that sets the rules for a subsequent economic interaction. This is different to the notion of institutions as actors, e.g. a university as an institution. In this sense, institutions themselves would not seek to improve mechanisms for information storage and retrieval. Rather, the actors in the political interaction (e.g. chiefs) would seek to do this, so they can better shape the rules of the economic interaction to benefit themselves.

We agree with this point, and have modified one sentence in the subsection "Role of Population" to avoid such anthropomorphization.

SI page 1: "at most one NGA controlling the polity at any time", the orders of NGA and polity should be switched here.

Thank you for catching this error, which we have fixed.